# Targeted Control of Gene Expression Using CRISPR-Associated Endoribonucleases

**DOI:** 10.3390/cells14070543

**Published:** 2025-04-03

**Authors:** Sagar J. Parikh, Heather M. Terron, Luke A. Burgard, Derek S. Maranan, Dylan D. Butler, Abigail Wiseman, Frank M. LaFerla, Shelley Lane, Malcolm A. Leissring

**Affiliations:** 1Institute for Memory Impairments and Neurological Disorders, University of California, Irvine, CA 92697, USA; 2Department of Neurobiology and Behavior, University of California, Irvine, CA 92697, USA

**Keywords:** CRISPR, direct repeat, endoribonuclease, gene regulation, RNA interference

## Abstract

CRISPR-associated endoribonucleases (Cas RNases) cleave single-stranded RNA in a highly sequence-specific manner by recognizing and binding to short RNA sequences known as direct repeats (DRs). Here, we investigate the potential of exploiting Cas RNases for the regulation of target genes with one or more DRs introduced into the 3′ untranslated region, an approach we refer to as DREDGE (direct repeat-enabled downregulation of gene expression). The DNase-dead version of Cas12a (dCas12a) was identified as the most efficient among five different Cas RNases tested and was subsequently evaluated in doxycycline-regulatable systems targeting either stably expressed fluorescent proteins or an endogenous gene. DREDGE performed superbly in stable cell lines, resulting in up to 90% downregulation with rapid onset, notably in a fully reversible and highly selective manner. Successful control of an endogenous gene with DREDGE was demonstrated in two formats, including one wherein both the DR and the transgene driving expression of dCas12a were introduced in one step by CRISPR-Cas. Our results establish DREDGE as an effective method for regulating gene expression in a targeted, highly selective, and fully reversible manner, with several advantages over existing technologies.

## 1. Introduction

CRISPR (clustered regularly interspaced short palindromic repeats) refers to a hallmark DNA sequence that plays a key role in antiviral defense in prokaryotes [1,2]. Repeated elements, known as direct repeats (DRs), flank interspaced sequences, known as spacers, that are perfectly complementary to the sequences of viral genomes [3]. In response to viral infection, the CRISPR region is transcribed to generate a pre-crRNA, which is subsequently processed by CRISPR-associated (Cas) single-stranded endoribonucleases (RNases) to excise the spacer RNAs [2]. Cas RNases carry out pre-crRNA processing by recognizing and binding to cognate DRs in a highly sequence-specific manner, then cleaving the RNA within or adjacent to the DR sequence [2]. The excised spacer RNAs, also known as guide RNAs (gRNAs), in turn, associate with Cas double-stranded deoxyribonucleases (DNases) to target and cleave specific DNA sequences within the invading viral genome [1].

Relative to Cas DNases, which have been widely exploited for the development of sophisticated methods for modifying genomic DNA and other applications [4], Cas RNases have received considerably less attention, with only a small number of studies investigating their potential for gene regulation [5,6,7,8,9,10], and even then often as an adjunct to conventional CRISPR-Cas technology [11]. Our group has been developing and evaluating methods for achieving drug-inducible downregulation of genes, ultimately for in vivo applications that, in the ideal case, are completely reversible and perfectly selective. We previously developed a method predicated on doxycycline (Dox)-dependent histone methylation of the promoter region of a target gene, which successfully downregulates a target gene at very low concentrations of Dox and was found to be fully reversible in cell culture after one week of Dox exposure [12]. However, accruing evidence indicates that the Krüppel-associate box repressor (KRAB) domain used in this system can disrupt gene expression irreversibly in a subset of cells after prolonged exposure to Dox [13,14,15,16], a possibility we explicitly aim to avoid, particularly in vivo.

We hypothesized that DRs and their cognate Cas RNases could be utilized as a powerful means for downregulating target genes selectively and fully reversibly, an approach we refer to as DREDGE (direct repeat-enabled downregulation of gene expression). Specifically, by introducing one or more DRs into the transcribed portion of a target gene, a cognate Cas RNase can be directed to cleave the modified mRNA specifically and exclusively, leading to its subsequent destruction by any of several mechanisms. While multiple instantiations are possible, in this study, we evaluate the approach of introducing one or more DRs into the 3′ untranslated region (UTR) of a target gene (Figure 1A). In this approach, dubbed 3′ *DREDGE*, a Cas RNase disrupts the expression of the target gene by catalyzing the removal of the poly(A) tail from the target gene mRNA, which triggers its rapid destruction via deadenylation-dependent mRNA decay [17] (Figure 1A). Relative to other approaches, there is a singular practical advantage to targeting the 3′ UTR: it allows for the possibility of introducing both the DR(s) (typically placed immediately downstream of the stop codon) and a transgene (TG) expressing the cognate RNase (inserted downstream of the 3′ end of the 3′ UTR) in one step via CRISPR-Cas—a significant time- and cost-saving measure, particularly for in vivo applications—which we implement herein.

The present study evaluates for the first time, to our knowledge, the feasibility of exploiting the RNase activity of Cas12a (also known as Cpf1)—specifically the DNase-dead version (dCas12a)—to downregulate the expression of target genes. This choice was inspired by a study showing that Cas12a completely eliminated the expression of a control protein whose transcript contained Cas12a DRs in its 3′ UTR [18]. A major goal of the latter study was to overcome this effect so that Cas12a and its crRNA could be expressed on a single transcript to, in one application, drive histone methylation via CRISPR interference (CRISPRi) using a single expression construct [18]. Ironically, the magnitude of the downregulation achieved with the latter approach in no case matched the essentially complete downregulation observed with the control vector featuring Cas12a DRs in its 3′ UTR [18], hence inspiring us to conceive of 3′ DREDGE.

We report here multiple successful implementations of 3′ DREDGE, including the fully reversible downregulation of the endogenous gene, *CTSD*, encoding cathepsin D (CatD), a lysosomal protease that our group has shown plays a critical role in the etiology of Alzheimer’s disease [19,20,21]. In a transient transfection paradigm, dCas12a and four different Cas RNases each markedly downregulated the expression of destabilized green fluorescent protein (GFPd2), with dCas12a markedly outperforming the other Cas RNases, achieving essentially 100% downregulation. In stable cell lines expressing GFPd2 constitutively and dCas12a in a Dox-dependent manner, DREDGE achieved > 90% downregulation with one or three Cas12 DRs and, moreover, exhibited full reversibility with a half-life (t_1/2_) of less than one day. RNA-seq analyses establish that DREDGE is highly selective for the target gene. We also utilized DREDGE to take control of the endogenous gene, *CTSD*, using CRISPR-Cas to introduce either three DRs alone or one DR together with a TG expressing dCas12a in a Dox-dependent manner. Both approaches resulted in efficient downregulation of *CTSD*—crucially—in a manner that was fully reversible after >2 months’ continuous exposure to Dox and also after repeated cycles of addition and withdrawal of Dox. Our results validate DREDGE as an effective means for downregulating target genes with full reversibility and—importantly—complete selectivity, circumventing off-target effects that commonly plague other approaches such as RNA interference (RNAi) and CRISPRi [22,23]. By virtue of these characteristics, DREDGE is expected to have broad utility for assessing biological responses to myriad transient phenomena ranging from injury to toxin exposure to drug treatments.

## 2. Materials and Methods

### 2.1. DNA Constructs

All constructs were assembled using the HiFi DNA Assembly Kit according to the manufacturer’s recommendations (New England Biolabs (NEB), Beverly, MA, USA) from DNA fragments generated either by restriction digestion, by PCR with Q5^®^ DNA Polymerase (NEB, Beverly, MA, USA), or by de novo DNA synthesis (Integrated DNA Technologies (IDT), San Diego, CA, USA). All constructs were verified by next-generation DNA sequencing (Azenta Life Sciences, Burlington, MA, USA). Source vectors included Addgene plasmids #14760 [24], #155307 [25], #183956 [26], #52119 (gift of Daria Vignali), #183962 [26], #111596 [27], and pTet-One [28] (Takara Bio USA, Inc., San Jose, CA, USA). Detailed cloning methods are provided in the Appendix A.

### 2.2. Cell Culture

All experiments employed wild-type, SV40-immortalized mouse embryonic fibroblasts (MEFs; Cat #CRL-2907; ATCC, Manassas, VA, USA). Cells were maintained at 37 °C in a humidified incubator supplemented with 5% CO_2_ in DMEM containing GlutaMAX^®^ supplemented with 10% Tet System Approved Fetal Bovine Serum (Takara Bio USA, San Jose, CA, USA), 100 U/mL penicillin, and 100 µg/mL streptomycin.

### 2.3. Flow Cytometry and FACS

For the screening of Cas RNases, MEFs were transfected with expression vectors using the Amaxa Nucleofector II electroporation system according to the manufacturer’s recommendation (Lonza Bioscience, Bend, OR, USA) and cultured overnight. Cells were then imaged on a Zeiss Axiovert 200 m fluorescent microscope (Zeiss, Dublin, CA, USA), harvested by trypsinization, and collected in tubes via a cell strainer. Flow cytometry on these cells and subsequently generated stable cell lines was performed on a BD LSRFortessa™ X-20 Cell Analyzer using lasers and filter sets optimized for the detection of GFP and mCherry fluorescence according to manufacturer’s recommendations (BD BioSciences, Franklin Lakes, NJ, USA). For the generation of stable cell lines, individual cells were plated into 96-well plates by fluorescence-activated cell sorting (FACS) using a BD FACSAria™ II Cell Sorter (BD BioSciences, Franklin Lakes, NJ, USA).

### 2.4. mRNA Collection and Analyses

Total mRNA from double-stable cell lines homogenized with QIAshredder™ was purified using RNeasy Plus Mini Kit, according to the manufacturer’s recommendations (QIAGEN, LLC, Germantown, MD, USA). Quantitative real-time PCR (RT-PCR) was performed on a CFX96 Touch Real-Time PCR Detection System using reagents from the SingleShot™ SYBR^®^ Green One-Step Kit according to the manufacturer’s recommendations (Bio-Rad, Hercules, CA, USA). Primer pairs (Fwd, Rev) were: GFP-A (ATGTCTTGTGCCCAGGAGAG, GTGGTATTTGTGAGCCAGGG); GFP-B (TGCTGGAGTTCGTGACCG, GCCGCCTACACATTGATCCT); and GAPDH (CCCACTCTTCCACCTTCGAT, GAGTTGGGATAGGGCCTCTC). Libraries for RNA-seq were prepared using the NEBNext^®^ Ultra™ II DNA Library Prep Kit for Illumina according to the manufacturer’s recommendations (NEB, Beverly, MA, USA). Next-generation sequencing was performed on a NovaSeq 6000 Sequencing System (Illumina, Inc., San Diego, CA, USA), and data were processed and analyzed by Azenta Life Sciences (Burlington, MA, USA).

### 2.5. Targeted Modification of the CTSD Locus

Three Cas12a DRs were introduced into the endogenous *CTSD* locus of MEFs by transfecting cells as above with a mixture comprising (1) recombinant Alt-R A.s. Cas12a (Cpf1) *Ultra* Nuclease; (2) two synthetic gRNAs (gRNA_2 and gRNA_3; see Appendix A); and (3) 0.5 µg of a PCR product of the targeted insert generated using phosphorothioate bond-containing oligonucleotides (IDT, San Diego, CA, USA). The DR + TG modification was done similarly, using Cpf1 Nuclease complexed to three synthetic gRNAs—gRNA_3, gRNA_9 (see Appendix A), and Syn_gRNA (GCTGTCCCCAGTGCATATTC) [29]—in this case using 2 µg of pCTSD_DR + TG (see Appendix A).

### 2.6. CatD Activity Assays

CatD activity assays were performed essentially as described [12,19] using the internally quenched, fluorogenic peptide substrate (Mca-GKPILFFRLK(Dnp)-R-NH_2_) (InnoPep, Inc. San Diego, CA, USA). In a typical reaction, near-confluent monolayers of MEFs were detached by brief digestion with trypsin-EDTA. After 2 washes in PBS, pelleted cells were lysed in 200 µL Lysis Buffer (40 mM NaOAc, 0.1% CHAPS, pH 3.5), incubated on ice for 10 min, then centrifuged at 21,000× *g* for 1 min. The concentration of the supernatant was estimated by A280 quantification using a NanoDrop^®^ ND-1000 UV-Vis Spectrophotometer (Thermo Fisher Scientific, Waltham, MA, USA). Samples were loaded into black 384-well plates, and reactions were initiated by the addition of an equal volume of Reaction Buffer (100 mM NaOAc, 0.2 M NaCl, pH 3.5) containing fluorogenic substrate (2 µM). Plates were immediately loaded into a microplate reader (Gemini EM, Molecular Devices, LLC, San Jose, CA, USA), and fluorescence (λ_ex_ = 328 nm, λ_em_ = 393 nm) was read continuously every 15 s for ≥10 min. Proteolytic activity was determined from initial slopes of progress curves from each well, obtained using SoftMax Pro (v. 5.0; Molecular Devices, LLC, San Jose, CA, USA), normalized to individual A280 readings and appropriate controls.

## 3. Results

### 3.1. Selection and Screening of Candidate Cas RNases

To assess the feasibility of downregulating target genes using the 3′ DREDGE approach (Figure 1A), we selected five different Cas RNases for investigation based on different criteria (Figure 1B). Cas12a (also known as Cpf1; Figure 1B), our initial and primary candidate for reasons outlined above, is unusual in possessing both DNase and RNase activity; consequently, we utilized a DNase-dead version of the protein, specifically the engineered “hyperdCas12a” DNase-dead version from *Lachnospiraceae bacterium* developed by Guo and colleagues [26]. Further bolstering our assessment that Cas12a, in particular, might constitute an especially effective Cas RNase, Magnusson and colleagues designed a short A/U-rich “synthetic separator” (synSeparator; Figure 1B)—AAAU—that enhances the excision of spacer sequences when positioned adjacent to the 5′ end of the Cas12a DR [30], which we incorporated into all Cas12a DR constructs (Figure 1B). Two other Cas RNases, PfCas6 and SsoCas6 (Figure 1B)—from *Pyrococcus furiosus* and *Sulfolobus solfataricus*, respectively—were selected based on reports that they are “multiple-turnover enzymes,” as distinct from single-turnover Cas RNases, which remain tightly bound to the cognate DR after cleavage [2]. Finally, CasE (also known as EcoCas6e) and Csy4 (also known as Cas6f)—from *Escherichia coli* and *Pseudomonas aeruginosa*, respectively—were selected based on a study showing these to be the best-performing of nine Cas RNases tested when introduced into mRNAs [18].

**Figure 1 cells-14-00543-f001:**
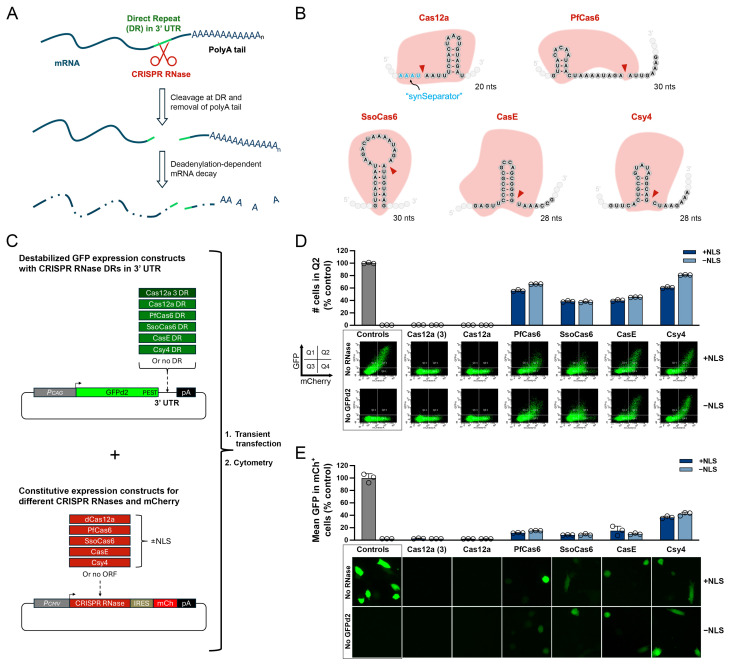
Screening of candidate Cas RNases for 3′ DREDGE. (**A**) Mechanism of 3′ DREDGE. Cleavage of the DR(s) in the 3′ UTR of mRNA (green) by a Cas RNase (red) removes the poly(A) tail, triggering rapid degradation. (**B**) DRs for the five Cas RNases investigated in this study, with cleavage sites indicated (red arrows). Note the “synSeparator” adjacent to the 5′ end of the Cas12a DR. (**C**) Designs of constructs expressing GFPd2 with different DRs (or no DR) in the 3′ UTR (green) and constructs expressing individual Cas RNases (or no RNase) together with mCherry (red). (**D**) Cell cytometry data for MEFs expressing different Cas RNases and GFPd2 constructs with their cognate DRs. Graph of the percentage of cells in Q2 relative to controls (top) within log-log plots of GFP vs. mCherry RFU values (n = 3 replicates), with representative plots shown (bottom). (**E**) Graph of GFP intensity in mCherry+ cells derived from RFU plots shown in (**D**) and normalized to controls, along with representative images of GFP fluorescence in cells prior to cytometry (bottom).

As illustrated in Figure 1B, the cognate DRs of the five Cas RNases contain certain commonalities and some key differences [11,31,32,33]. The DRs are all quite short, comprising ≤30 nucleotides, but highly varied in primary nucleotide sequence (Figure 1B). The DR for Cas12a is the shortest, comprising just 20 nucleotides (neglecting the four-nucleotide synSeparator included in all constructs) (Figure 1B). A second distinguishing feature is the placement of the cleavage sites. All the tested Cas RNases except Cas12a cleave within the DR, at a position seven or eight nucleotides from the 3′ end; Cas12a, by contrast, cleaves outside the DR, at the 5′ end (Figure 1B). Finally, the five DRs differ in the extent to which they form hairpins and the degree of hydrogen bonding within each hairpin, with CasE featuring a hairpin comprised of six G/C base pairs [11] and PfCas6 featuring a hairpin comprised of just three A/U base pairs [31].

To compare the relative efficacy of the five selected Cas RNases for downregulating a target gene, we generated a two-part model system that uses GFP fluorescence as a convenient marker of target gene expression and mCherry as a marker of Cas RNase expression (Figure 1C). The first part of this system consisted of vectors expressing a destabilized form of GFP (GFPd2), featuring a very short, two-hour half-life (t_1/2_) [24], driven by the strong CAG promoter. We cloned one DR for each of the five Cas RNases into the 3′ UTR of this construct, or no DR as a control (Figure 1C, top). For dCas12a, we also tested a three-DR version, wherein the DRs flank two “dummy” spacers with no complementarity to mouse genomic DNA [30] (Figure 1C, top; see Figure 4B). For the second part of this system, each of the five Cas RNases was cloned into a vector co-expressing mCherry (Figure 1C, bottom). For each Cas RNase, we generated versions either lacking or containing a nuclear localization signal (NLS) (two in the case of Cas12a) for a total of 10 Cas RNase expression constructs plus an empty-vector, “No-RNase” control (Figure 1C).

Each Cas RNase/mCherry expression construct was cotransfected together with the GFPd2 construct bearing the cognate DR(s) in its 3′ UTR into mouse embryonic fibroblasts (MEFs). Controls consisted of: (1) MEFs transfected with empty (No-RNase) mCherry vector plus GFPd2 lacking a DR (No-DR), representing the maximum possible GFP fluorescence; and (2) cells transfected with empty mCherry vector alone (No-GFPd2), representing the minimum possible GFP fluorescence (Figure 1D). One day after transfection, cells were harvested and analyzed by FACS. Log-log plots of the fluorescence in each cell were generated, depicting GFP and mCherry on the Y- and X-axes, respectively, which were further divided into four quadrants by using untransfected MEFs as a non-fluorescent control to establish strict boundaries for GFP and mCherry fluorescence (Figure 1D). In control cells cotransfected with No-DR GFPd2 and No-RNase mCherry, abundant GFP and mCherry fluorescence were present, resulting in large numbers of cells appearing in the upper-right quadrant (Q2) of RFU plots; conversely, for cells expressing mCherry alone, no cells were present in Q2, as expected (Figure 1D). Relative to No-RNase controls, cells expressing each of the five Cas RNases and their cognate DRs all showed decreases in the percentage of cells in Q2 to varying extents (Figure 1D), together with substantial reductions in the mean level of GFP fluorescence in mCherry-positive cells (e.g., cells in Q2 and Q4; Figure 1E). Significantly, dCas12a performed superiorly relative to all other Cas RNases tested by both metrics, with almost no cells appearing in Q2 (Figure 1D) and GFP fluorescence levels being indistinguishable from No-GFPd2 controls (Figure 1E). There were no detectable differences between GFPd2 constructs expressing one or three Cas12a DRs and dCas12a in this experimental paradigm (Figure 1D,E).

The absence or presence of an NLS on the Cas RNases only modestly impacted their performance (Figure 1D,E). Marginal decreases in efficacy were observed for a few Cas RNases lacking an NLS, although for others, the NLS had either no effect or the opposite effect (Figure 1D,E). Nevertheless, this parameter was sufficiently important to us to warrant additional testing. Consequently, we generated an additional dCas12a/mCherry expression construct containing a nuclear exclusion sequence (NES), which was compared in parallel with the dCas12a constructs lacking or possessing two NLS sequences in cells co-expressing GFPd2 with one Cas12a DR (or no DR) in its 3′ UTR. Consistent with the previous results, no significant differences were observed between the constructs containing an NES, an NLS, or neither localization sequence (Appendix A). Nevertheless, in light of the modest improvements in performance observed for some RNases possessing NLSs, we elected to proceed with dCas12a constructs containing two NLS sequences.

To verify that the observed downregulation was specifically attributable to the interaction between dCas12a and its cognate DR, we performed additional control experiments. First, we showed that the downregulation of GFPd2 with one DR by dCas12a can be completely rescued by co-transfection with a GFPd2 expression construct lacking a DR (Figure 2A,B). Second, we showed that GFP downregulation by dCas12a was achieved only for the GFPd2 construct containing its cognate DR and not for constructs containing the DR from a different Cas RNase, SsoCas6 (Figure 2C,D). The same was true for SsoCas6: GFP expression was downregulated only in the presence of constructs containing its cognate DR and not in those containing the DR for Cas12a (Figure 2C,D).

### 3.2. Dox-Regulatable Gene Expression by 3′ DREDGE Using dCas12a

We next sought to characterize the performance of cells expressing dCas12a in a doxycycline (Dox)-dependent manner, using the Tet-One™ system [28] (Takara Bio USA, Inc., San Jose, CA, USA). To that end, we first created cell lines stably expressing GFPd2 with zero, one, or three Cas12a DRs (Figure 3A). These stable cell lines were subsequently used to generate double-stable lines also expressing dCas12a (with two NLSs) together with mCherry (with three NLSs) and neomycin resistance, all under the control of the TRE promoter to permit Dox-regulatable expression (Figure 3B). The latter construct lacking dCas12a served as a No-RNase control. As illustrated in Figure 3C,D, all control lines performed as anticipated. For the No-DR GFPd2 cells lines, essentially 0% and 100% of cells appeared in Q2 in the absence or presence of Dox, respectively, whether expressing dCas12a or no RNase (Figure 3C, left), and mean GFP levels were essentially unchanged, irrespective of RNase expression or Dox administration (Figure 3D, left). Cells stably expressing GFPd2 with either one or three Cas12a DRs in the 3′ UTR behaved similarly in the presence of Dox (1 µg/mL) when co-expressing the No-RNase Dox-inducible control vector (Figure 3C,D, middle and right). In striking contrast, Dox-induced expression of dCas12a in GFPd2 lines with one or three DRs elicited marked reductions in the number of cells in Q2 (89.4% and 84.8%, respectively) relative to the same parental lines expressing No-RNase mCherry controls (Figure 3C, middle and right). Similarly, relative to cells without Dox treatment, cells with Dox-induced dCas12a expression exhibited 93.2% and 93.7% reductions in mean GFP fluorescence in lines with one and three Cas12a DRs, respectively (Figure 3D, middle and right).

To more completely characterize the performance of 3′ DREDGE, we used these stable lines to carry out Dox dose-response experiments (Figure 3E,F) as well as time courses of the responsiveness of gene expression after the addition or removal of Dox (Figure 3G). Dose-response experiments conducted in the cell lines with one and three DRs revealed IC_50_s of 110 and 204 ng/mL Dox, respectively, using the percent of mCherry-positive cells (Q2 + Q4) present in Q2 as a metric (Figure 3E). Similar results were obtained using mean GFP fluorescence in all cells, yielding IC_50_s of 187 and 307 ng/mL Dox, respectively (Figure 3F). Notably, the mean mCherry fluorescence (representing the average in all runs in both cell lines) exhibited a similar EC_50_ of 260 ng/mL Dox (Figure 3F), suggesting that the percent downregulation of GFPd2 was essentially a direct reflection of dCas12a expression. Significantly, time courses revealed that the 3′ DREDGE approach exhibits remarkably fast kinetics, with t_1/2_s for induction of downregulation of 0.52 and 0.81 d for one and three DRs, respectively (Figure 3G). The t_1/2_s for restoration of activity were similar: 0.61 and 0.95 d, respectively. Overall, both the one-DR and the three-DR systems performed comparably; however, it is noteworthy that the one-DR system consistently performed marginally better on all the foregoing measures (see Section 4).

### 3.3. 3′ DREDGE Regulates the mRNA of Target Genes with High Selectivity

3′ DREDGE is postulated to downregulate the expression of target genes via selective removal of the poly(A) tail from the target mRNA, which in turn is well established to result in rapid destruction of the mRNA by deadenylation-dependent decay [17]. To verify that this mechanism is operative, we quantified poly(A) mRNA derived from our double-stable cell lines using RT-PCR. GFPd2 mRNA levels were first compared in cell lines expressing dCas12a or No RNase, derived from the same stable cell line expressing GFPd2 with one Cas12a DR. Using two separate primer pairs (GFP-A, GFP-B), GFPd2 mRNA levels were found to be reduced ~90% in the line expressing dCas12a compared to the No-RNase control (Figure 4A), closely matching the relative reduction in GFPd2 fluorescence as quantified by cell cytometry (Figure 3C, middle). We then performed RT-PCR on the dCas12a-expressing version of the latter lines, in this case, in the absence or presence of Dox (1 µg/mL). We obtained a 98.3% reduction in GFPd2 mRNA levels (Figure 4B), again in good agreement with relative GFP fluorescence measured by cell cytometry (Figure 3D, middle).

**Figure 3 cells-14-00543-f003:**
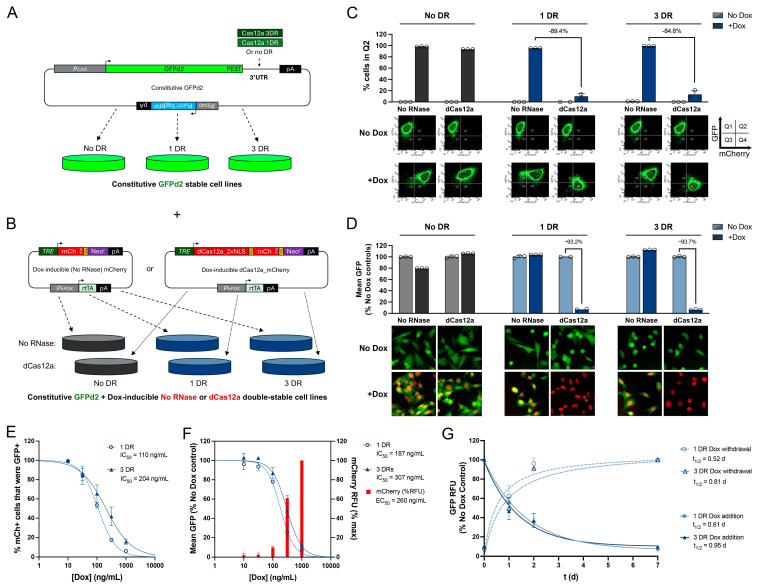
Dox-regulatable control of gene expression by 3′ DREDGE. (**A**) Design of constructs constitutively expressing GFPd2 with zero, one, or three Cas12a DRs in the 3′ UTR used to create three different stable cell lines. (**B**) Design of constructs with Dox-regulatable co-expression of dCas12a (or No RNase) and mCherry used to create double-stable cell lines from the lines in (**A**). (**C**) Percentage of cells in Q2 for double-stable cell lines expressing GFPd2 with zero, one, or three DRs and also conditionally expressing either Cas12a or no RNase, tested in the absence or presence of Dox (top) derived from log-log plots of GFP vs. mCherry RFU (bottom). (**D**) Mean GFP RFU in the cell lines in (**C**) in the absence or presence of Dox derived from cell cytometry (top) with representative images of cells in the different conditions (bottom). Data in (**C**,**D**) are normalized to Dox-treated No-DR and No-RNase controls; n = 2–3 per condition. (**E**,**F**) Dose-response curves showing (**E**) the percent of mCherry+ cells (i.e., Q2 + Q4) that were also GFP+ (i.e., in Q2) and (**F**) mean GFP RFU as a function of Dox dose in stable cell lines with one or three DRs conditionally expressing cDas12a. Red columns in (**F**) show the mean mCherry RFU as a function of Dox dose for both cell lines, normalized to the maximum at 1000 ng/mL. Mean IC_50_ values for all dose-response experiments are indicated. Data are mean ± SEM, normalized to values in the absence of Dox for each line; n = 2–3 per condition. (**G**) Time courses of GFP RFU in response to addition (solid lines) or withdrawal (dashed lines) of Dox in the cell lines in (**E**,**F**) normalized to No-Dox controls. Mean half-life values (t_1/2_) are shown. Data are mean ± SEM for 2–3 independent experiments.

A primary motivation for developing 3′ DREDGE was to create a method for downregulating gene expression in a manner that is highly selective for the targeted gene. It was, therefore, important to conduct a genome-scale analysis of mRNA expression to assay for possible effects on off-target gene expression. To that end, we performed RNA-seq on mRNA obtained from 4 different double-stable cell lines, each collected in triplicate: (1) GFPd2 with one DR expressing dCas12a, (2) GFPd2 with one DR expressing No RNase, (3) GFPd2 with no DR expressing dCas12a, and (4) GFPd2 with no DR expressing No RNase. After the collection of mRNA and generation of dual-indexed cDNA libraries, the 12 libraries were combined, and next-generation sequencing was performed, yielding >400 million reads (Appendix A), which were subsequently analyzed for differential gene expression in pairwise comparisons (Appendix A). As shown in Figure 4C, when the Cas12a DR was present, GFPd2 mRNA was downregulated in dCas12a-expressing cells relative to No RNase-expressing cells in a remarkably selective manner. The magnitude of the log_2_ fold-decrease (−2.40) was essentially the same as that obtained by RT-PCR (−3.04 and −2.99, cf., Figure 4A), and the adjusted *p*-value for the difference was 6.7 × 10^−280^, a value hundreds of orders of magnitude more significant than other differentially expressed genes (DEGs; Figure 4C). As summarized in Figure 4D, other pairwise comparisons between the different cell lines yielded essentially similar results, with significant reductions in GFPd2 mRNA expression occurring only when dCas12a is expressed in the presence of one DR; conversely, no significant differences in expression were observed for pairwise comparisons that did not include this particular line (see Appendix A). Also noteworthy was the relatively low percentage of DEGs in all comparisons (Figure 4D). These cell lines had each been passaged at least 12 times since being derived from common lines and were frozen prior to resurrection for these experiments, so some proportion of DEGs was entirely expected. Nevertheless, the absolute percentages of DEGs were not only relatively low (e.g., 1.9% in for the comparison in Figure 4C), but there was no pattern in their relative values consistent with an effect of dCas12a (or other variables) on the number of DEGs. These results are also in agreement with other bioinformatic evidence. In particular, a BLASTN search performed with the Cas12a DR sequence revealed no matches with more than 13 consecutive nucleotides within the NCBI RefSeq mRNA sequences in the mouse genome (Appendix A). Moreover, the top hits from this search were not among the DEGs in any of the pairwise comparisons (Appendix A). Collectively, these results strongly suggest that 3′ DREDGE mediated by dCas12a is highly selective for targeted genes.

**Figure 4 cells-14-00543-f004:**
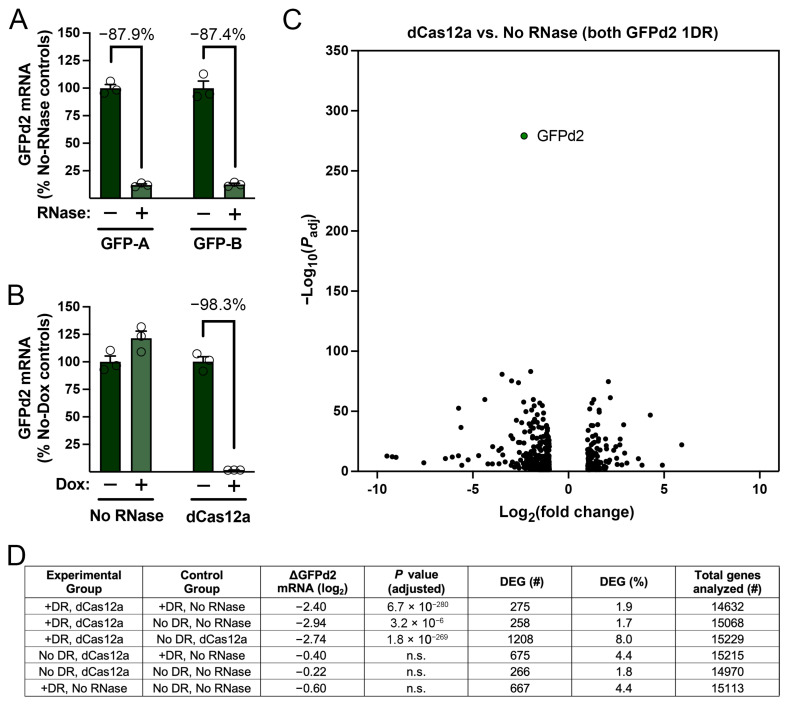
Analysis of mRNA expression in 3′ DREDGE by RT-PCR and RNA-seq. (**A**) RT-PCR results for mRNA obtained from cell lines expressing GFPd2 with one DR in the 3′ UTR and either No RNase (−) or dCas12a (+), evaluated using two different primer pairs specific for GFPd2 mRNA (GFP-A, GFP-B). (**B**) RT-PCR results for the latter dCas12a-expressing line, grown in the absence (−) or presence (+) of Dox (1 µg/mL), using GFP-A primers. Data for (**A**,**B**) are mean ± SEM for three biological replicates, each representing the average of three individual RT-PCR reads. (**C**) Volcano plot showing differentially expressed genes (DEG) between cell lines expressing GFPd2 with one DR together with either dCas12a or the No RNase control. The results for the GFPd2 mRNA are labeled and highlighted (green circle). Additional results can be found in Appendix A, and all RNA-seq analyses can be accessed online at https://osf.io/kumjr (accessed on 1 April 2025). (**D**) Table summarizing the results from all pairwise comparisons. The number of DEGs represents genes with *p* values smaller than a Bonferroni-corrected value unique to each comparison (i.e., *p* < 0.05 divided by the total number of genes analyzed). Note that GFPd2 mRNA was significantly downregulated only in the cell line expressing both dCas12a and GFPd2 with one DR.

### 3.4. Dox-Regulatable Expression of the Endogenous Gene, CTSD, by 3′ DREDGE

We next sought to assess the feasibility of downregulating an endogenous gene with the 3′ DREDGE approach. Having previously developed an alternative method for downregulating *CTSD* [12], we selected this gene as our target. We tested two basic configurations (Figure 5 and Figure 6). In the first, we used CRISPR-Cpf1 in MEFs to introduce three Cas12a DRs within the 3′ UTR of *CTSD* (Figure 5A; Appendix A). To this end, we used a gene-trap approach with a knockin construct featuring an internal ribosomal entry site (IRES) driving expression of a puromycin resistance gene (Puro^r^) fused to GFP (both flanked by LoxP sites) followed by three Cas12a DRs, all flanked by ~600-bp 5′ and 3′ homology arms (Figure 5A; Appendix A). This construct lacked a promoter and also a poly(A) tail, so it could express Puro^r^ if and only if it integrated successfully into the 3′ UTR of *CTSD*. After puromycin selection, we identified individual positive clones by PCR, including one line featuring one copy of the integrated construct and one *CTSD* allele inactivated by non-homologous end-joining (NHEJ; Appendix A), which was used for all downstream analyses. After removal of the floxed IRES_Puro^r^::GFP elements with Cre-recombinase, the modified CTSD allele contained only the three Cas12a DRs (plus an upstream LoxP site) (Figure 5B; Appendix A).

**Figure 5 cells-14-00543-f005:**
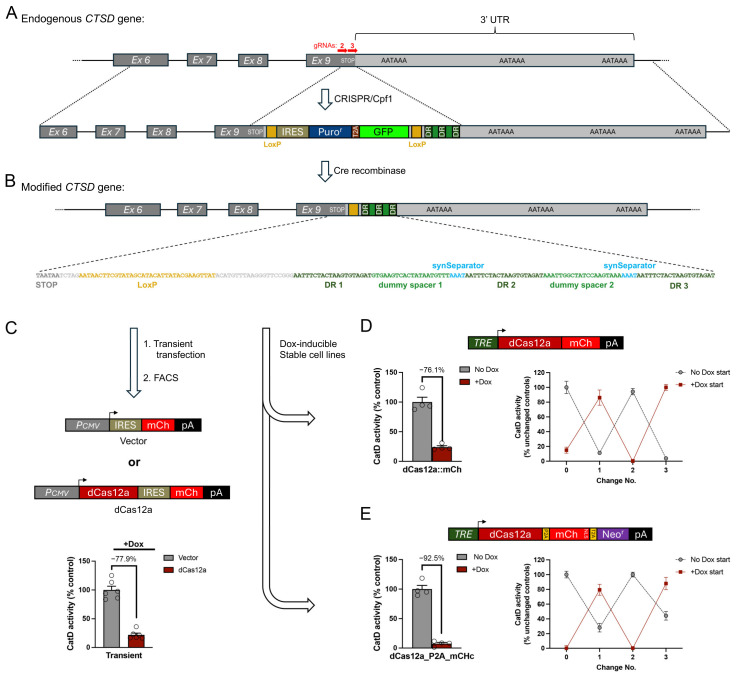
Dox-regulatable control of endogenous *CTSD* expression by 3′ DREDGE. (**A**) Genomic structure of the 3′ end of murine *CTSD* (top) and design of the “gene-trap” targeting construct used to insert 3 Cas12a DRs into the 3′ UTR by CRIPSR/Cpf1 (bottom). Note the absence of a poly(A) signal within the targeting construct and the presence of upstream IRES and LoxP sites flanking all of the inserted elements except the 3 Cas12a DRs. (**B**) Design of the modified *CTSD* gene after targeted insertion of the gene-trap construct and removal of the IRES and Puro^r^-T2A-GFP ORF by Cre-recombinase. The sequence of the inserted region, beginning at the stop codon in Exon 9 is shown. (**C**) Overview of the design of DNA constructs used for transient transfection experiments (top) and the outcome of CatD activity assays performed on mCherry+ cells collected 24 h later by FACS (bottom). Data are mean ± SEM; n = 6. (**D**,**E**) Stable cell lines created with a 3DR-containing parental cell line based on two Dox-regulatable constructs expressing a dCas12a::mCherry fusion (**D**, top) or the construct described in Figure 2B, co-expressing dCas12a, mCherry, and Neo^r^ from a single transcript via 2A peptides (**E**, top) Shown are Dox-dependent downregulation of CatD proteolytic activity achieved in the initial characterization both constructs (**D**,**E**, bottom left) and after repeated changes from +Dox to no Dox and vice versa spaced ~1 week apart (**D**,**E**, bottom right). Data are mean ± SEM; n = 4–8 replicates per condition.

To assess the ability of this modified allele to effect the downregulation of CatD, we transiently transfected this cell line with either a constitutive dCas12a/mCherry expression vector or an empty (No-RNase) mCherry-only vector (Figure 5C). One day later, we collected mCherry-positive cells from both conditions by FACS and conducted CatD activity assays (see Section 2), which revealed that CatD levels in dCas12a-expressing cells were reduced by 77.9% relative to cells transfected with the No RNase control (Figure 5C).

To generate stable clones, we transfected the modified three-DR cell line with Dox-inducible vectors expressing dCas12a of two different designs, both co-expressing mCherry either as a fusion with dCas12a (Figure 5D) or as a separate protein (via the incorporation of a P2A element; Figure 5E). After the addition of Dox to induce dCas12a expression, these stable cell lines exhibited marked reductions in CatD activity—of 76.1% and 92.5%, respectively—relative to the same cell lines maintained in the absence of Dox (Figure 5D,E, left). Importantly, CatD activity within these lines in the absence of Dox was comparable to control lines expressing No-RNase control vectors, whether treated with Dox or not, even after extensive incubation with Dox (for ~2 months) during the selection of these lines. Moreover, the two dCas12a-expressing lines both exhibited full reversibility even after being subjected to multiple successive alternating treatments without or with Dox (Figure 5D,E, right).

In the second configuration, we aimed to introduce both a single Cas12a DR within the 3′ UTR of *CTSD* (immediately downstream of the stop codon) and a complete transgene (TG) for Dox-inducible dCas12a expression (downstream of the 3′ end of the *CTSD* 3′UTR), a configuration we refer to as “DR + TG” (Figure 6A,B). Note that we included two FRT sites flanking the TG portion of the insert so that it could be removed by Flp recombinase if desired (Figure 6A). As before, CRISPR-Cpf1 was used to introduce the DR + TG construct into the endogenous *CTSD* gene, in this case using two gRNAs, one near the stop codon of the *CTSD* ORF and the other downstream of the 3′ end of the *CTSD* 3′ UTR (Figure 6A; Appendix A). For this construct, we also incorporated a third “synthetic gRNA” (Syn-gRNA) sequence outside of each ~600-bp homology arm, which enabled the DR + TG construct to be excised from the vector backbone simultaneously with CRISPR-Cas-mediated recombination, a technique shown to improve CRISPR efficiency [29]. Despite the presence of significant homology in the middle of our construct (~700 bp)—in the form of the complete 3′ UTR of CTSD (plus one DR)—we encountered no difficulty identifying a clone with one copy of the DR + TG construct successfully integrated (and the other allele inactivated by NHEJ) (Appendix A).

When CatD activity within this DR + TG cell line was quantified in the absence vs. the presence of Dox, CatD activity was significantly reduced by Dox, albeit only by 44.9% (Figure 6B, right). There are several different plausible explanations for this less-than-ideal outcome, ranging from the fundamental to the technical, and it was important to distinguish among them. On the fundamental side, it could be that a single copy of the dCas12a TG is simply insufficient to effectively downregulate CatD using the 3′ DREDGE method. Alternatively, perhaps a single DR might be insufficient to effect complete downregulation irrespective of dCas12a levels. On the technical side, it could be that an active *CTSD* allele might be present that escaped our detection or was operative despite the NHEJ deletion detected by PCR. Another plausible technical explanation pertained to the design of the Dox-inducible dCas12a TG. The DR + TG construct was designed and created early on in this study, well before we had the benefit of experience with alternative dCas12a transgene designs, and it contained a few sub-optimal features, including (1) a PEST sequence at the C-terminus of dCas12a (which we had introduced to promote rapid recovery of CatD activity after Dox withdrawal) (Figure 6B), (2) an intron after the TRE (which we had postulated would be beneficial for in vivo applications to promote mRNA processing) (Figure 6B), and (3) an ORF for dCas12a that was not codon optimized to ensure no cryptic splice sites or polyadenylation signals were present.

**Figure 6 cells-14-00543-f006:**
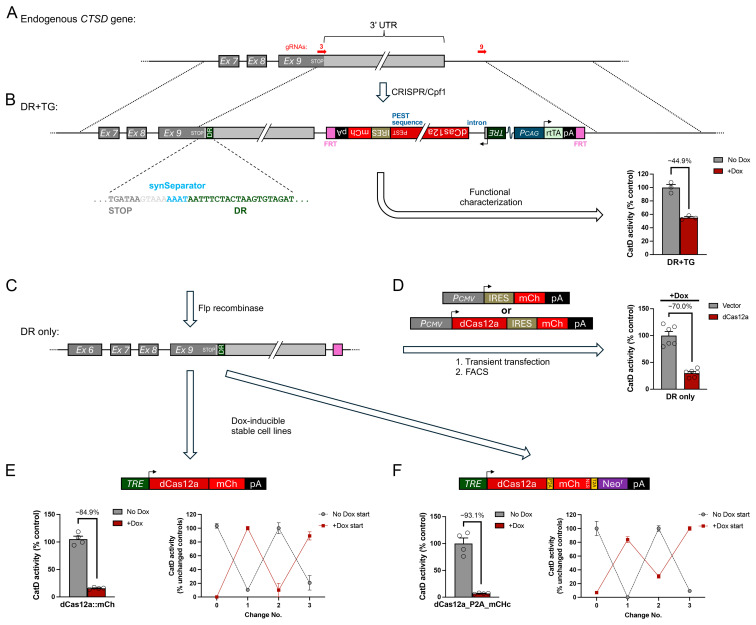
Dox-regulated control of endogenous *CTSD* expression using the “DR + TG” approach. (**A**) Genomic structure of the 3′ end of murine *CTSD*, indicating the locations of gRNAs used for CRISPR/Cpf1. (**B**) Design of the DR + TG insert used to introduce both a single dCas12a DR immediately downstream of the stop codon (sequence shown) and a complete transgene (TG) for Dox-regulatable expression of dCas12a flanked by FRT sites (purple). Note the inclusion of several design elements subsequently deemed to be suboptimal, including an artificial intron after the TRE and a C-terminal PEST sequence on dCas12a. Functional characterization of a line with one copy of the modified allele and one allele functionally knocked out by NHEJ (bottom right). Note that CatD activity was decreased by only ~45%. Data are mean ± SEM; n = 3 replicates per condition. (**C**) Design of the “DR-only” modified allele after removal of the TG with Flp-recombinase. (**D**) Design of DNA constructs used for transient transfection experiments (left) and the outcome of CatD activity assays performed on mCherry+ cells collected 24 h later by FACS (right). Data are mean ± SEM; n = 6. Note that larger reductions in CatD activity were achieved by this method. (**E**,**F**) Design and performance of stable cell lines generated from the DR-only cell line based on Dox-regulatable constructs expressing a dCas12a::mCherry fusion (**E**, top) or co-expressing dCas12a, mCherry, and Neo^r^ (**F**, top). Shown are Dox-dependent downregulation of CatD activity achieved in the initial characterization of both constructs (**E**,**F**, bottom left) and after repeated changes from +Dox to no Dox and vice versa spaced ~1 week apart (**E**,**F**, bottom right). Data are mean ± SEM; n = 4–8 replicates per condition.

To discriminate among these possibilities, we removed the dCas12a TG portion within the DR + TG line using Flp recombinase (Figure 6C; Appendix A). This “DR-only” line was then transiently transfected with a vector constitutively expressing dCas12a and mCherry (or mCherry-only vector), and after collection of mCherry-positive cells from both conditions, CatD activity was assessed in the presence of Dox (Figure 6D). CatD activity was decreased more completely in this paradigm (by 70.0%), suggesting that there was, in fact, a single functional *CTSD* allele in this line that could, in fact, be downregulated more completely (Figure 6D). To confirm this, we generated stable cell lines expressing the two versions of Dox-inducible dCas12a expression vectors (or No-RNase controls) tested previously in the three-DR cell line (Figure 6E,F). These cell lines behaved similarly to the three-DR stable lines, with dCas12a-expressing cells showing marked decreases in CatD activity of 84.9% and 93.1%, respectively, relative to the same lines grown in the absence of Dox. Moreover, as was the case for the three-DR cell lines, downregulation of CatD activity by both expression vectors in this one-DR line was fully reversible following multiple rounds of alternation between Dox addition and withdrawal (Figure 6E,F, right panels).

## 4. Discussion

Our results establish DREDGE as an attractive general approach for regulating gene expression with several distinct advantages over widely used alternative methods. First, DREDGE offers a high degree of selectivity for the targeted gene by virtue of the inclusion of one or more DRs as a *cis*-element within the targeted gene. Both RNAi and CRISPRi, by contrast, are known to have significant off-target effects that are difficult to predict and control for, owing to their reliance on RNA::RNA and RNA::DNA complementarity, respectively, which tolerate mismatches to varying degrees [22,23]. By contrast, RNA-seq-based evaluation of the DREDGE approach revealed that the target gene is very selectively downregulated, with no evidence of off-target effects attributable to the Cas RNase. Second, as we show, DREDGE can be used to control gene expression in a completely reversible manner, a distinct advantage vis-à-vis CRISPRi and other approaches dependent upon histone methylation, which have been documented to result in irreversible methylation in several systems [13,14,15,16]. Other techniques for reversibly disabling proteins, such as the “anchor away” approach [34,35], wherein binding of a drug to a fused domain excludes a target protein from the nucleus, or the auxin-inducible degron (AID) system [36], wherein binding of a drug triggers rapid degradation of proteins fused to a mini-AID tag, have disadvantages: the former is applicable only to a subset of target proteins operative in the nucleus, and both require the addition of fused tags to the target gene, which may or may not be tolerated. Third, DREDGE controls gene expression with very rapid kinetics, both for downregulation due to the high efficiency of deadenylation-dependent mRNA decay [17] and for recovery therefrom, as a consequence of targeting a constantly replenished pool of mRNA. These kinetic properties, in particular, should facilitate the study of the consequences of transient disruptions to gene expression, as can occur, for example, by exposure to discrete toxins [37]. Fourth, as implemented here in multiple configurations, DREDGE consistently resulted in a higher degree of downregulation (>80%) than is typical for RNAi and CRISPRi (<70%) [18,26,38], which can facilitate the study of phenotypes dependent upon relatively complete downregulation. Finally, DREDGE requires minimal modifications to the target gene (e.g., as small as 20 nucleotides in the case of the DR for Cas12a), which compares favorably to the size of other *cis*-elements utilized for reversible gene regulation (e.g., >250 nucleotides in the case of the TRE [28]). The very small size of DRs minimizes the chances of perturbing expression of the targeted gene by affecting mRNA secondary structure or disrupting possible regulatory elements, such as microRNA binding sites [39].

The 3′ DREDGE approach specifically evaluated in this study has further unique advantages vis-à-vis alternative implementations. For instance, targeting the 3′ UTR makes it possible to regulate genes with multiple isoforms due to alternative transcription start sites because such genes frequently share a common stop codon [40]. Moreover, as we demonstrated, targeting the 3′ UTR also makes it feasible to introduce both the DR and a transgene for a Cas RNase in one step using CRISPR-Cas. This feature of 3′ DREDGE makes it especially attractive for implementation in animal models, where single-locus modifications for gene regulation are strongly favored over the alternative of crossing lines with multiple modified genetic loci.

Given the complexity inherent in implementing site-directed genomic modifications, DREDGE is unlikely to constitute a primary substitute for more widely used methods of gene downregulation and may also have limited utility for translational applications, which would presumably require homozygous modifications to be effective. In our view, DREDGE is best suited for situations where a particular gene is being targeted, and one wishes to achieve highly selective downregulation of that gene while minimizing the *a priori* possibility of off-target effects, which is enhanced by the use of a *cis*-element not found elsewhere in the genome. Though it is not commonly done, if selectivity is truly paramount, approaches relying on DNA::DNA or DNA::RNA complementarity should, in principle, be rigorously assayed for off-target effects prior to implementation, particularly in the case of long-term projects such as animal modeling studies. Given the additional work inherent in confirming the selectivity of these more conventional methods, we feel DREDGE compares favorably despite the initial effort required for its implementation.

Our study also reveals potential limitations of 3′ DREDGE, at least for the specific implementation tested herein. In particular, continuous, very high levels of expression of Cas RNases appear to be required to efficiently downregulate target genes. In the example of our DR + TG approach, where just a single copy of the transgene was integrated, we obtained only partial (~45%) downregulation. Although our results suggest that these less-than-ideal results may be attributable to suboptimal features of the transgene design, they establish the point that, as implemented here with dCas12a, the efficiency of DREDGE is directly determined by Cas RNase expression levels. Since the expression of endogenous genes can vary greatly, DREDGE may not be suitable for genes expressed at particularly high levels (unless further optimized; see below). Fortunately, there are resources that estimate the numbers of mRNA molecules in different cell types and tissues [41], which may help assess the a priori suitability of DREDGE for specific genes.

To our knowledge, this is the first study to investigate the potential of Cas12a for regulating gene expression via its RNase activity. Recent studies offer insights into the RNase activity of Cas12a that are highly relevant to its performance in DREDGE. Detailed kinetic analyses reveal that Cas12a interacts with the pre-crRNA via exquisitely potent interactions, with equilibrium constants in the range of 56 pM to 3 nM for the RNA::protein complex, depending upon the species studied [42,43,44]. Given that RNA binding is the rate-limiting step in the processing of mRNAs by Cas12a [44], this is a favorable property. On the other hand, the half-life for dissociation of the Cas12a::pre-crRNA complex was recently reported to be 26 h [44], indicating that RNA binding is essentially irreversible and implying that only one mRNA molecule can be degraded by each Cas12a molecule. Fortunately, Cas12a cuts at the 5′ end of the DR (see Figure 1B), so it remains associated with the 3′ end of the cleaved mRNA containing the poly(A) tail and thus does not interfere with deadenylation-dependent decay of the remainder of the mRNA. However, the essentially irreversible binding of Cas12a to each cognate DR suggests that Cas12a might be more effective if implemented with a single DR rather than multiple DRs, a prediction that is supported by the results we obtained here for systems with one vs. three DRs (see Figure 3E,F).

Although Cas12a performed well in 3′ DREDGE, certain features of Cas12a could potentially be optimized to further improve its performance, though significant additional work would be required. For example, because it possesses both DNase and RNase activities, Cas12a is an especially large protein (~140 kDa); expression might be made more efficient if truncated versions of the protein containing only the RNase activity could be developed. Also, it seems conceivable that the RNA::protein interaction could plausibly be engineered to render this interaction reversible and thereby convert Cas12a to multiple-turnover Cas RNase [2]; this might be achieved by varying the sequence of the DR [45] and/or by manipulating resides within Cas12a involved in DR binding.

## 5. Conclusions

We conclude that DREDGE is an effective general method for regulating endogenous gene expression, particularly in circumstances where a high degree of selectivity and full reversibility are paramount, such as animal modeling studies. The 3′ DREDGE approach, in particular, also allows for a convenient method for rapid implementation of this system in one step via CRISPR-Cas, using the DR + TG approach; however, our results also illustrate that the effectiveness of Cas RNases expressed via a single copy of a transgene depends critically upon the relative expression level of the target gene and a suitably effective transgene design. Although our results establish dCas12a as an efficient mediator of 3′ DREDGE, the particular instantiation of this approach tested here is unlikely to be the most efficient of all possible configurations. The general approach might be improved by further engineering of dCas12a or, potentially, by the discovery of alternative Cas RNases with higher RNA processing efficiency. We hope our findings will stimulate further improvements to this fundamental approach for targeted control of gene expression.

## Figures and Tables

**Figure 2 cells-14-00543-f002:**
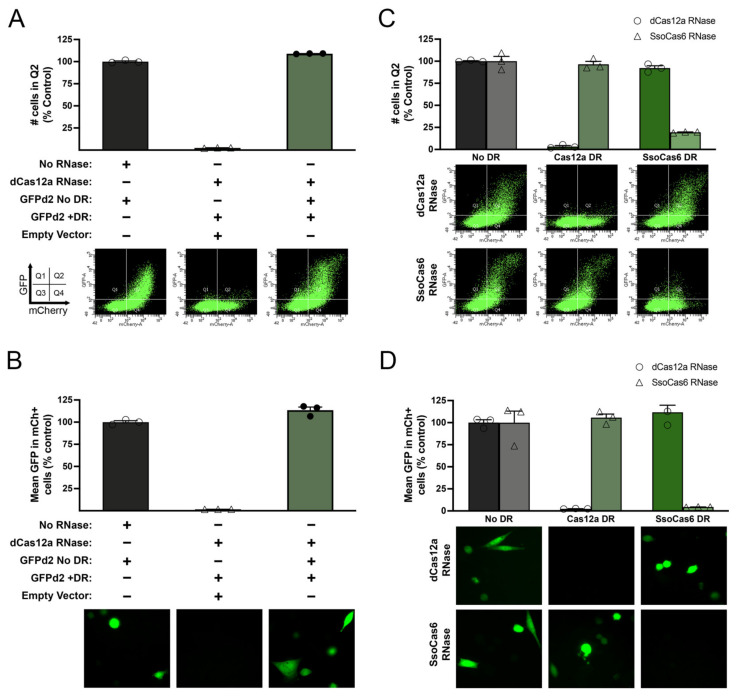
3′ DREDGE is mediated by specific interactions between the Cas RNase and its cognate DR. (**A**,**B**) Downregulation of the expression of GFPd2 with one Cas12a DR by dCas12a can be completely rescued by the addition of GFPd2 lacking a DR, as assessed both by the percent of cells within Q2 (**A**) and by mean GFP fluorescence (**B**). (**C**,**D**) Downregulation by dCas12a (○) or SsoCas6 (△) occurs only for GFPd2 constructs containing the cognate DR and not for constructs containing the DR for the other Cas RNase, as assessed both by the percent of cells within Q2 (**C**) and by mean GFP fluorescence (**D**). Data are mean ± SEM normalized to No-DR controls, n = 3.

## Data Availability

Data are contained within the article and Appendix A. Original, raw fluorescence microscopy image files and complete RNA-seq analyses are deposited at the Open Science Framework repository, available at: www.doi.org/10.17605/OSF.IO/KUMJR (accessed on 1 April 2025).

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
