# Peer review of "Targeted Control of Gene Expression Using CRISPR-Associated Endoribonucleases"

_cells, 2025, doi:10.3390/cells14070543_

Round 1

Reviewer 1 Report

Comments and Suggestions for Authors

The manuscript presents a novel and potentially valuable approach to gene expression regulation using CRISPR-associated RNases (DREDGE). The study is well-designed and provides extensive experimental validation. The method offers a highly selective and reversible alternative to existing gene regulation techniques. The authors thoroughly evaluate multiple Cas RNases, demonstrating their comparative effectiveness. The study includes strong data support, reinforcing the approach’s reliability.

However, the manuscript is difficult to read due to excessive technical detail, particularly in describing basic molecular biology methods such as restriction sites and plasmid constructs. While important, these details detract from readability and should be moved to Supplementary Methods. Additionally, the system’s complexity raises concerns about its ease of implementation compared to more straightforward gene knockdown techniques. A clearer discussion of the feasibility of broader applications would strengthen the manuscript.

I recommend acceptance with minor revisions to improve clarity. To do this, I recommend streamlining technical descriptions and shifting excessive methodological details to supplementary materials.

Author Response

We thank this Reviewer and all others for their helpful comments.  We performed several, key experiments in response, including RT-PCR, rescue experiments and cognate vs. non-cognate to establish the specificity of the RNases for the DRs, and—most significantly—RNA-seq analysis on 4 different cell lines to assay for potential off-target effects of 3' DREDGE.  These experiments resulted in the addition of 5 new figures, including 2 in the main manuscript and 3 in the Supplemental Materials.  These additions have greatly strengthened the manuscript, particularly in terms of establishing the selectivity of 3' DREDGE for control of target genes, and we are very grateful to the Reviewers for all their input.

Our responses to each individual suggestion are summarized, below.

Comment 1: The manuscript presents a novel and potentially valuable approach to gene expression regulation using CRISPR-associated RNases (DREDGE). The study is well-designed and provides extensive experimental validation. The method offers a highly selective and reversible alternative to existing gene regulation techniques. The authors thoroughly evaluate multiple Cas RNases, demonstrating their comparative effectiveness. The study includes strong data support, reinforcing the approach’s reliability.

Response 1:  We thank the reviewer for the kind remarks.

Comment 2: However, the manuscript is difficult to read due to excessive technical detail, particularly in describing basic molecular biology methods such as restriction sites and plasmid constructs. While important, these details detract from readability and should be moved to Supplementary Methods. Additionally, the system’s complexity raises concerns about its ease of implementation compared to more straightforward gene knockdown techniques. A clearer discussion of the feasibility of broader applications would strengthen the manuscript.

Response 2: We agree that too much technical detail was included—indeed, we were surprised that the template for this journal placed the methods in the second section, prior to the results section.  As suggested, we have moved the bulk of the methods to a new Supplementary Methods section within the Supplemental Information.  While we do agree that this method is certainly more complex than other gene knockdown techniques, we only resorted to it out of dissatisfaction with the latter.  With RNAi and CRISPRi, we have been unable to downregulate CTSD as completely as needed for our purposes.  Moreover, both of these methods, while certainly more straightforward, have issues with selectivity and the latter method with complete reversibility.  As we emphasize throughout the manuscript, and now highlight in a new paragraph added to the Discussion (Lines 560-572), this novel approach is best suited for circumstances where both high selectivity and complete reversibility are paramount, particularly for in vivo applications, which is our main purpose for developing this technology. As we point out, if selectivity is really a primary goal, then exhaustive genome-scale analyses ought to be conducted prior to implementing approaches using either RNAi or CRISPRi, a quantity of work that is multiplied if several targeting sequences need to be compared.  Our approach requires more initial effort vis-à-vis these methods, to be sure, but only one design needs to be implemented, and our RNA-seq data strongly support the high degree of selectivity of this approach.  For those of us creating animal models—in which near-complete downregulation and complete reversibility of target genes are mandatory—the development of DREDGE has allowed us to overcome limitations of other approaches, and we feel this technology will be of value to other groups encountering similar difficulties. 

Comment 3: I recommend acceptance with minor revisions to improve clarity. To do this, I recommend streamlining technical descriptions and shifting excessive methodological details to supplementary materials.

Response 3: We thank the Reviewer for the recommendation and the suggestions, which we feel have greatly improved the manuscript.

Reviewer 2 Report

Comments and Suggestions for Authors

The manuscript by Parikh et al. describes the protocol for efficient regulation of gene expression using CRISPR endoribonuclease or Cas RNase, which is the DNase-dead version of Cas12a. The method involves incorporation of one or more DRs in the 3’UTR of target gene. Authors propose that the conditional expression of Cas12a results in selective cleavage of DR in the 3’UTR of mRNA of target gene. The net result is removal of the poly(A) tail of mRNA, triggering rapid degradation of mRNA, thereby downregulating expression of the target gene. Authors demonstrate reversible regulation of two genes, GFP and cathepsin D, using this system, which they refer to as DREDGE. The approach is novel as it uses Cas RNase activity to regulate a target gene at the posttranscriptional level. Furthermore, their data shows that the system is fully reversible. The approach has the potential to reversibly regulate expression of any gene in mammalian cell lines in a highly selective manner. The manuscript can be improved further by addressing following issues:

  1. Authors claim that DREDGE approach regulates expression of genes at the level of mRNA stability by selectively degrading mRNA of target gene. They, however, provide no data in support of their argument. To complete the study, authors must show that the level of GFP and cathepsin D mRNA decreases upon expression of Cas 12a. They can do so by measuring the level of nascent mRNA by nuclear run-on assay as well as steady state mRNA level of GFP and cathepsin D before and after expression of Cas 12a. If authors hypothesis is correct, the level nascent mRNA will remain unaffected, but the steady state mRNA level will decrease upon regulated expression of Cas 12a.
  2. Authors claim that, unlike RNAi and CRISPRi, the DREDGE approach has no significant off target effects. There is, however, no data to substantiate their claim. Authors need to provide experimental evidence to show that DREDGE has no off-target effects.
  3. In the discussion section, authors must compare DREDGE approach with other similar approaches like anchor away approach and auxin-inducible degron (AID) approach, which is very widely used nowadays in diverse biological systems.

Author Response

We thank this Reviewer and all others for their helpful comments.  We performed several, key experiments in response, including RT-PCR, rescue experiments and cognate vs. non-cognate to establish the specificity of the RNases for the DRs, and—most significantly—RNA-seq analysis on 4 different cell lines to assay for potential off-target effects of 3' DREDGE.  These experiments resulted in the addition of 5 new figures, including 2 in the main manuscript and 3 in the Supplemental Materials.  These additions have greatly strengthened the manuscript, particularly in terms of establishing the selectivity of 3' DREDGE for control of target genes, and we are very grateful to the Reviewers for all their input.

Reviewer Comment: The manuscript by Parikh et al. describes the protocol for efficient regulation of gene expression using CRISPR endoribonuclease or Cas RNase, which is the DNase-dead version of Cas12a. The method involves incorporation of one or more DRs in the 3’UTR of target gene. Authors propose that the conditional expression of Cas12a results in selective cleavage of DR in the 3’UTR of mRNA of target gene. The net result is removal of the poly(A) tail of mRNA, triggering rapid degradation of mRNA, thereby downregulating expression of the target gene. Authors demonstrate reversible regulation of two genes, GFP and cathepsin D, using this system, which they refer to as DREDGE. The approach is novel as it uses Cas RNase activity to regulate a target gene at the posttranscriptional level. Furthermore, their data shows that the system is fully reversible. The approach has the potential to reversibly regulate expression of any gene in mammalian cell lines in a highly selective manner. The manuscript can be improved further by addressing following issues:

Reply: We appreciate this Reviewer's kind remarks.

Comment 1: Authors claim that DREDGE approach regulates expression of genes at the level of mRNA stability by selectively degrading mRNA of target gene. They, however, provide no data in support of their argument. To complete the study, authors must show that the level of GFP and cathepsin D mRNA decreases upon expression of Cas 12a. They can do so by measuring the level of nascent mRNA by nuclear run-on assay as well as steady state mRNA level of GFP and cathepsin D before and after expression of Cas 12a. If authors hypothesis is correct, the level nascent mRNA will remain unaffected, but the steady state mRNA level will decrease upon regulated expression of Cas 12a.

Reply 1:  We are very appreciative of these suggestions, which we have successfully addressed in multiple ways.  Although removal of the poly(A) tail is well-established to trigger deadenylation-dependent mRNA decay, as we cited in the paper, there are indeed alternative mechanisms that may have been operative (and in other contexts, we have in fact encountered this).  To address this, we performed RT-PCR in two key comparisons, one in which the same parental cell line containing one Cas12a DR in the 3' UTR of the GFPd2 mRNA was expressing either dCas12a or no RNase, and another in which the former cell line expressing dCas12a in a Dox-dependent manner was evaluated in the absence versus the presence of Dox.  Consistent with the hypothesized mechanism, GFPd2 mRNA was downregulated by >~85% in the first comparison, as evaluated using two separate primers targeting GFP, and GFPd2 mRNA was downregulated by >98% in the second comparison.  In addition, based on suggestion 2, as detailed below, we performed RNA-seq experiment confirming that (A) the DREDGE approach is highly selective for the target gene and (2) the degree of downregulation was consistent with that observed by RT-PCR. These results have been included as a new Figure 4.

Comment 2: Authors claim that, unlike RNAi and CRISPRi, the DREDGE approach has no significant off target effects. There is, however, no data to substantiate their claim. Authors need to provide experimental evidence to show that DREDGE has no off-target effects.

Reply 2: This was an excellent suggestion, which we addressed by performed RNA-seq.  Briefly, we obtained mRNA from 4 different cell lines in triplicate, expressing GFPd2 with or without a Cas12a DR and expressing either dCas12a or no RNase, performing next-generation sequencing on dual-indexed cDNA libraries that yielded >400 million reads.  As shown in the new Figure 4C and 4D (and Supplemental Figure S3), we obtained truly remarkable confirmation not only that the DREDGE approach is highly selective for the target gene, but also that there was no pattern of differential gene expression corresponding to the presence vs. the absence of the dCas12a expression or the DR. These results have been included as a new Figure 4.  Finally, we now include a BLASTN search performed using the Cas12a DR, which we show exhibits no complementarity exceeding 13 consecutive nucleotides in any mRNA within the mouse genome.  The genes that do show this low degree of complementarity, moreover, do not appear among the differentially expressed genes detected by RNA-seq.  The results of this search have been provided in new Supplemental Figures S4.

Comment 3: In the discussion section, authors must compare DREDGE approach with other similar approaches like anchor away approach and auxin-inducible degron (AID) approach, which is very widely used nowadays in diverse biological systems.

Reply 3:  We thank the reviewer for this suggestion.  We have added a discussion of these systems to the first paragraph of the Discussion section (Lines 530-536), including primary citations for the two techniques (which were very cool).

Reviewer 3 Report

Comments and Suggestions for Authors

The manuscript’s findings have important implications for researchers who need precise, reversible control of gene expression. This “3′ DREDGE” approach offers an innovative angel by exploiting the RNase activity of Cas enzymes to degrade mRNAs, the incorporation of minimal DR sequences for specificity and the potential for one-step gene knock-in with co-integration of the Cas RNase transgene adds novelty to the current field of CRISPR-based gene editing technique.

However, improvements need to be made in:

  1. Study design: The authors included several appropriate controls: “no-DR” (no direct repeat); “no-RNase” constructs, to demonstrate that the knockdown is dependent on the presence of both the direct repeat and the Cas endoribonuclease. However, off‑target Controls is missing, it would be essential to include a “scrambled” or “non‑cognate” direct‑repeat sequence as a control—one that theoretically should not be cleaved by the chosen Cas RNase.
  2. A genetic “rescue” experiment (e.g., reintroducing the gene without the DR) is necessary to support statement in summary. It could further confirm specificity and rule out unintended global effects of Cas RNase overexpression.
  3. Method Section: Please provide brief rationales for using particular restriction sites and mention any issues of large insert stability.
  4. A more thorough evaluation of off-target effects is missing from the Methods section. The authors should incorporate transcriptomic profiling or a suitable global screening approach to confirm the absence of unintended targets. Otherwise, the possibility of off-target effects should be clearly acknowledged as a limitation in this study’s discussion.
  5. As a novel approach, the translational value of this work (what is the next step on optimizing this system) needs to be discussed thoroughly in the discussion part.

Author Response

We thank this Reviewer and all others for their helpful comments.  We performed several, key experiments in response, including RT-PCR, rescue experiments and cognate vs. non-cognate to establish the specificity of the RNases for the DRs, and—most significantly—RNA-seq analysis on 4 different cell lines to assay for potential off-target effects of 3' DREDGE.  These experiments resulted in the addition of 5 new figures, including 2 in the main manuscript and 3 in the Supplemental Materials.  These additions have greatly strengthened the manuscript, particularly in terms of establishing the selectivity of 3' DREDGE for control of target genes, and we are very grateful to the Reviewers for all their input.

Comment 1: The manuscript’s findings have important implications for researchers who need precise, reversible control of gene expression. This “3′ DREDGE” approach offers an innovative angel by exploiting the RNase activity of Cas enzymes to degrade mRNAs, the incorporation of minimal DR sequences for specificity and the potential for one-step gene knock-in with co-integration of the Cas RNase transgene adds novelty to the current field of CRISPR-based gene editing technique.

Reply 1: We thank this reviewer for the kind remarks.

Comment 2: However, improvements need to be made in:

Study design: The authors included several appropriate controls: “no-DR” (no direct repeat); “no-RNase” constructs, to demonstrate that the knockdown is dependent on the presence of both the direct repeat and the Cas endoribonuclease. However, off‑target Controls is missing, it would be essential to include a “scrambled” or “non‑cognate” direct‑repeat sequence as a control—one that theoretically should not be cleaved by the chosen Cas RNase.

Reply 2: We thank the reviewer for this suggestion, which has been addressed in a set of experiments now presented in a new Figure 2.  To address the specificity for the cognate vs. a non cognate DR, we performed transient transfection experiments with two Cas RNases, dCas12a and SSoCas6, and GFPd2 constructs containing either no DR or the DR for Cas12a and SSoCas6.  Consistent with their being a selective interaction, downregulation was achieved only in the case where the Cas RNase and the cognate DR were expressed.  Referring to the suggestion to test a "scrambled" version of the DR, we checked the GFPd2 sequence for just such a sequence (wherein the same number of As, Ts, Gs and Cs was present within a given window) and identified one matching the Cas12a DR that was present in all GFPd2 mRNAs (present at 994 nucleotides downstream of the initiation codon).  This happenstance—and the above experiments—confirms the well-established fact that Cas RNases interact with their cognate DRs in a highly sequence-specific manner (see DeWeirdt et al., Nat Biotechnol 39, no. 1 (2021): 94-104.).  These results appear in our new Figure 2 (panels C and D).

Comment 3: A genetic “rescue” experiment (e.g., reintroducing the gene without the DR) is necessary to support statement in summary. It could further confirm specificity and rule out unintended global effects of Cas RNase overexpression.

Reply 3: This was a great suggestion.  Again using a transient transfection approach, we show that the addition of a GFPd2 construct lacking a DR rescues the downregulation of cells co-expressing a GFPd2 construct with a Cas 12a DR and the dCas12a RNase.  These results appear in the new Figure 2 (panels A and B).

Comment 4: Method Section: Please provide brief rationales for using particular restriction sites and mention any issues of large insert stability.

Reply 4: Another Reviewer felt strongly that the methods for the cloning were overly detailed.  This being the sole concern of this reviewer, we feel it would be going in the wrong direction to explain why particular restriction sites were used (which is in any event a matter of convenience in almost every case).  Note that we also moved the detailed cloning methods to the Supplemental Information at the behest of this Reviewer.  Although the cloning of very large constructs required the GateWay system, we encountered no issues of large insert stability, provided we amplified constructs using recA-negative bacteria, such as NEBstables.

Comment 5: A more thorough evaluation of off-target effects is missing from the Methods section. The authors should incorporate transcriptomic profiling or a suitable global screening approach to confirm the absence of unintended targets. Otherwise, the possibility of off-target effects should be clearly acknowledged as a limitation in this study’s discussion.

Reply 5: This was an excellent suggestion, also made by another Reviewer, which we addressed by performed RNA-seq. Briefly, we obtained mRNA from 4 different cell lines in triplicate, expressing GFPd2 with or without a Cas12a DR and expressing either dCas12a or no RNase.  As shown in the new Figure 4C and 4D (and Supplemental Figure S3), we obtained truly remarkable confirmation not only that the DREDGE approach is highly selective for the target gene, but also that there was no pattern of differential gene expression corresponding to the presence vs. the absence of the dCas12a expression or the DR. These results have been included within a new Figure 4 (Panels C and D) and also in Supp. Figs. S2 and S3.

Comment 6: As a novel approach, the translational value of this work (what is the next step on optimizing this system) needs to be discussed thoroughly in the discussion part.

Reply 6: We have addressed our assessment of the translational value of this work within a new paragraph added to the Discussion section (Lines 560-572).  We already included a discussion of possible ways to optimize this system, including the suggestion of testing alternate Cas RNases, of modifying Cas12a in ways that would make it smaller, and in modifying either Cas12a or the Cas12a DR in a way that rendered the RNase a multiple turnover enzyme, rather than a single-cut enzyme.  

Round 2

Reviewer 2 Report

Comments and Suggestions for Authors

Authors have taken care of all my concerns.

Reviewer 3 Report

Comments and Suggestions for Authors

The authors addressed the scientific points; please carefully proofread the English.